# Sugar Beet Pulp as a Biorefinery Substrate for Designing Feed

**DOI:** 10.3390/molecules28052064

**Published:** 2023-02-22

**Authors:** Dawid Dygas, Dorota Kręgiel, Joanna Berłowska

**Affiliations:** Department of Environmental Biotechnology, Lodz University of Technology, 171/173 Wólczańska Street, 90-530 Łódź, Poland

**Keywords:** waste biomass, sugar beet pulp, single cell protein, yeast, fiber, protein, hydrolysates

## Abstract

An example of the implementation of the principles of the circular economy is the use of sugar beet pulp as animal feed. Here, we investigate the possible use of yeast strains to enrich waste biomass in single-cell protein (SCP). The strains were evaluated for yeast growth (pour plate method), protein increment (Kjeldahl method), assimilation of free amino nitrogen (FAN), and reduction of crude fiber content. All the tested strains were able to grow on hydrolyzed sugar beet pulp-based medium. The greatest increases in protein content were observed for *Candida utilis* LOCK0021 and *Saccharomyces cerevisiae* Ethanol Red (ΔN = 2.33%) on fresh sugar beet pulp, and for *Scheffersomyces stipitis* NCYC1541 (ΔN = 3.04%) on dried sugar beet pulp. All the strains assimilated FAN from the culture medium. The largest reductions in the crude fiber content of the biomass were recorded for *Saccharomyces cerevisiae* Ethanol Red (Δ = 10.89%) on fresh sugar beet pulp and *Candida utilis* LOCK0021 (Δ = 15.05%) on dried sugar beet pulp. The results show that sugar beet pulp provides an excellent matrix for SCP and feed production.

## 1. Introduction

One of the 17 Sustainable Development Goals is to ensure sustainable consumption and production patterns, which can be achieved through wider application of the principles of the circular economy [1]. In line with these priorities, the sugar industry in Poland is striving to generate as little waste as possible, by reusing technological tools (water, heat) or creating new products based on waste (bioethanol, biogas) [2,3]. Globally, the main raw material for the sugar industry is sugar cane. However, due to climate conditions, European sugar factories use sugar beet [4]. The European Union is the world’s largest producer of beet sugar, accounting for around 50% of global production. According to 2021 data on sugar beet pulp exports, the largest producers of sugar beets are Russia (USD 240 million, 1.2 million metric tonnes), Egypt and the Arab Republics (USD 80 million, 306,000 metric tonnes), and the USA (USD 73 million, 301,000 metric tonnes) [5]. Most sugar beet produced by the EU is grown in the northern part of Europe, where the climatic conditions are most favorable. The most competitive production areas are in northern France, Germany, the Netherlands, Belgium, and Poland. In Poland each year 1.5 million tonnes of sugar beet are processed, resulting in 0.2 million tonnes of sugar beet pulp with 20% dry mass. Such huge quantities of biomass require thoughtful design and implementation of waste management systems. Possible ways of managing sugar beet pulp waste are presented in Figure 1.

Sugar beet pulp is characterized by 18–23% dry mass content [8], but contains relatively large amounts of protein (11.5–20.25%) and crude fiber (20.7%) [9,10]. This waste material has a wide range of potential applications, spanning the biofuel industry, renewable energy, the production of platform chemical reagents, and animal feed [11,12,13]. Recently, there has been increasing interest in the creation of biorefinery conglomerates combining multiple types of companies, capable of processing biomass and further waste biomass into new high value products [14]. By creating relationships with sugar beet farmers, buying raw material from farmers, revalorizing waste biomass, and delivering it to livestock farmers, it is possible to implement a conglomerate model.

Currently, most sugar beet pulp is used as a feedstuff for dairy cows under different feeding regimens [15]. Usually, it is used as roughage to supplement feed requirements by volume, or as an element in total mixed ration feeding [16]. Sugar beet pulp waste can also be modified to increase its nutritional value. One way to the enrich the nutritional value of sugar beet pulp is to use it as a carbon source for the cultivation of microorganisms rich in single-cell protein (SCP). Plant biomass liquefaction can be conducted in separate hydrolysis and fermentation (SHF) mode or by simultaneous saccharification and fermentation (SSF). Often, pre-hydrolysis is required for proper process initialization [17]. As a result of enzyme activity, the plant polysaccharides are depolymerized. The active enzymes gradually release the saccharides, providing suitable conditions for yeast growth [17,18].

The aim of the present study was to evaluate different yeast strains as potential producers of SCP from sugar beet pulp. In this way, more valuable feed can be obtained, contributing to the goals of sustainable consumption and production, especially in terms of sustainable management and efficient use of natural resources (12th Sustainable Development Goal). The presented solution could bring greater economic benefits with minor changes in distribution channels. Sugar factories are often bound by permanent contracts with farmers who deliver beets and collect the pulp.

## 2. Results

### 2.1. Determination of Carbohydrates

The characterization and quantification of sugar content is a crucial step for assessing the suitability of a substrate for yeast growth [19]. The sugar beet pulp as a raw material is composed mainly of polysaccharides, such as 22–24 wt.% cellulose, 30 wt.% hemicelluloses and 15–25 wt.% pectin, along with small amounts of fat, protein, ash and lignin [20]. The polymeric fractions may be decomposed enzymatically. By comparing the concentrations of sugars in biomass samples before and after pre-hydrolysis, it is also possible to obtain a realistic estimate of the environmental conditions needed for yeast multiplication and of the potential of the carbon sources available. The content of sugars in samples subjected to enzymatic hydrolysis thus enables prediction of the effectiveness of enzyme preparations applied to the biomass. Carbohydrates primarily used as carbon sources for microbial growth by yeast were selected [7,18]. The sugar profiles of the two kinds of sugar beet pulp before and after enzymatic treatment are presented in Table 1.

Enzymatic hydrolysis of crude sugar beet pulp resulted in an increase in all monosaccharides in all the tested biomasses, depending on the enzyme dose. The largest increase of 260–340 times was detected for glucose, with concentrations of 13.03–17.28 g/L after hydrolysis, in comparison to 0.05 g/L for the non-hydrolyzed sample in both kinds of biomass. The smallest increase in sugar concentration was noted for mannose, which increased to 0.02–0.65 g/L from 0.01 g/L in the control sample.

### 2.2. Yeast Growth

Ten strains of yeast belonging to both conventional and unconventional strains were used in the study. The strains were chosen for their ability to utilize the compounds present in the obtained hydrolysates (Table 1), based on the authors’ experience from previous experimental work [21,22]. Yeast multiplication was evaluated based on the plate count method. This enabled us to compare the ability of cells to adapt and grow in the conditions created by the fresh and dried sugar beet pulp pretreated by enzymatic hydrolysis with various doses of enzymes (Table 2 and Table 3).

The largest number of yeast cells, 2.82 × 10^8^ CFU/mL, was recorded for *Scheffersomyces stipitis* in the presence of sugar beet pulp pre-hydrolyzed with an enzyme dose of 0.5 mL/10 g DM. Reducing the dose of enzyme used for hydrolysis resulted in lower sugar content and reduced yeast growth. The lowest cell multiplication was recorded for *Yarrowia lipolytica* strain (1.32 × 10^7^ CFU/mL) cultivated with sugar beet pulp at an enzyme dose of 0.25 mL/10 g DM. 

Similar analyses of yeast multiplication were performed for dried sugar beet pulp (Table 3).

All tested yeast strains showed ability to grow in culture media with dried sugar beet pulp. The highest cell count was recorded for the *Saccharomyces cerevisiae* TT strain (2.04 × 10^8^ CFU/mL) in medium with an enzyme dose of 0.125 mL/10 g DM. The lowest yeast growth with the same enzyme dose was noted for *Candida utilis* R6. For samples treated with an enzyme dose of 0.25 mL/10 g DM, the highest number of cells was obtained for *Saccharomyces cerevisiae* Ethanol Red (1.64 × 10^8^ CFU/mL) and the lowest for *Candida utilis* R6 (2.40 × 10^7^ CFU/mL). The highest enzyme dose of 0.5 mL/10 g DM used for biomass pretreatment created an environment suitable for growth of *Yarrowia lipolytica* (1.36 × 10^8^ CFU/mL). The lowest yield in this medium was noted for *Candida utilis* R7 (5.67 × 10^7^ CFU/mL).

### 2.3. Protein Content

The ability of yeast strains to bioconvert substrates into the organic form of microbial protein is the main parameter for assessing their suitability for the revalorization of different types of waste biomass [23,24]. The results of enrichment of sugar beet pulp with yeast protein are presented in Figure 2.

The increase in protein content in relation to the raw material was evaluated using the Kjeldahl method. The protein increment in the yeast samples ranged from 0.57% to 2.33%. The greatest increase was noted in the samples after hydrolysis with the highest enzyme dose. The best enrichment after cultivation on fresh sugar beet pulp pre-hydrolyzed with enzymes at a dose of 0.5 mL/10 g DM was noted for *Candida utilis* strains and *Saccharomyces cerevisiae* Ethanol Red. The lowest increase was noted for the *Metschnikowia pulcherrima* strain (1.69%). Reducing the enzyme dose resulted in a concomitant reduction in the protein increment of the sample. For example, the protein increment for samples supplemented with the enzyme at a dose of 0.25 mL/10 g DM was lower and ranged from 0.79% to 2.11%. The highest increment under the same conditions was recorded for *Candida utilis*. The lowest was measured for the *Yarrowia lipolytica* strain. A further reduction in the concentration of the enzyme portion to 0.125 mL/10 g DM resulted in a reduction in protein gain efficiency. The lowest rise in this case was 0.57% for *Candida utilis* R6. The highest rise was 1.67% for *Candida utilis* LOCK0021.

The protein content was also analyzed for yeasts cultivated in dried sugar beet (Figure 3).

All tested yeast strains showed great potential as SCP producers. The highest protein yield obtained after yeast cultivation with dried sugar beet pulp hydrolyzed by the enzymes at a dose of 0.5 mL/10g DM was noted for *Scheffersomyces stipitis* (3.04%). The lowest protein yield was noted for *Candida utilis* R6 (1.62%). In general, lowering the enzyme dose resulted in a reduction in protein yield. However, there were some exceptions. For a concentration of 0.25 mL/10 g DM, the ΔN increased from 1.00% (*Metschnikowia pulcherrima*) to 2.06% (*Saccharomyces cerevisiae* Ethanol Red). The lowest enzyme dose (0.125 mL/10 g DM) resulted in the highest ΔN for cultures of *Scheffersomyces stipitis* (2.41%) and was least beneficial for *Kluyveromyces marxianus* (0.82%). Therefore, differences in yeast multiplication in culture media containing dried sugar beet biomass were reflected in differences in protein content.

### 2.4. Free Amino Nitrogen (FAN) Content in Fresh and Dried Sugar Beet Pulp Samples

After yeast multiplication in culture media based on sugar beet pulp, the amounts of free amino nitrogen (FAN) in the fresh and dried sugar beet samples were determined. Hydrolyzed samples, not subjected to fermentation, were used as control samples (Figure 4 and Figure 5).

Analysis before and after yeast cultivation on fresh and dried sugar beet pulp, showed a decrease in FAN from 231.97–276.85 mg/L to 38.21–157.52 mg/L after fermentation. The obtained results allowed to determine a statistically significant difference in all tested samples compared to the control samples. All strains successively assimilated FAN from the culture medium and bioconverted it to the organic form of proteins. This confirms that sugar beet pulp is a good yeast substrate.

### 2.5. Crude Fiber Content

The crude fiber content of waste determines its potential technological uses. Biomass must meet certain requirements to be considered as an animal feed or feed component. Biomass samples after cultivation with the selected yeast strains were analyzed for crude fiber content. Strains characterized by the highest protein increases were chosen (Figure 6 and Figure 7).

A reduction in crude fiber content was noted for all samples of fresh sugar beet pulp after enzyme treatment in the pre-hydrolysis step. Reducing the dosage of the enzyme preparation negatively affected the efficiency of fiber reduction. The fiber content after fermentation ranged from 11.51% to 21.78%, compared to 21.97% and 22.4% for the hydrolyzed biomass and the raw material, respectively. The lowest fiber content was recorded for the *Saccharomyces cerevisiae* Ethanol Red strain, with an enzyme dose of 0.5 mL/10 g DM. The fiber content was 11.51%. The lowest crude fiber reduction capacity for this enzyme concentration was observed for the *Candida utilis* R6 strain. The fiber content was 15.00%. An enzyme dose of 0.25 mL/10 g DM allowed a fiber content ranging from 13.3% for *Scheffersomyces stipitis* NCYC1541 strain to 17.54% for *Saccharomyces cerevisiae* strain Ethanol Red. Reducing the enzyme dose to 0.125 mL/10 g DM resulted in the smallest reduction in crude fiber content of the biomass. The measured fiber content ranged from 18.08% to 21.78%. The highest crude fiber reduction efficiency was observed for the *Candida utilis* R7 strain, and the lowest for *Candida utilis* LOCK0021.

The fiber content of the dried sugar beet pulp also showed a reduction in relation to unfermented raw material. Enzymatic hydrolysis of the biomass at a dose of 0.5 mL/10 g DM followed by fermentation with the *Candida utilis* strain resulted in the greatest reduction, to 13.96%. The least efficient strain at the same enzyme dose was *Saccharomyces cerevisiae* Ethanol Red, with a reduction to 19.34%. Reducing the dose to 0.12 mL/10 g DM resulted in an increase in the relative crude fiber content of the biomass. The lowest content was recorded for the *Candida utilis* strain, at 15.68%, and the highest at 22.36% for *Kluyveromyces marxianus*. A further reduction in enzyme dose resulted in an increase in the fiber content of the biomass. At an enzyme concentration of 0.125 mL/10 g DM, the lowest fiber content was recorded for *Scheffersomyces stipitis* NCYC1541 strain at 17.55% and the highest at 27.25% for *Saccharomyces cerevisiae* Ethanol Red.

## 3. Discussion

Reducing the dose of enzyme used for hydrolysis resulted in a concomitant reduction in the concentration of sugars obtained after the process. Possible discrepancies may be due to the uneven distribution of the enzyme preparation in the full volume of the sample. Interestingly, enzymatic hydrolysis of fresh sugar beet pulp resulted in higher concentrations of sugars compared to the hydrolysate of dried waste material. This may be a result of the better action of enzymes in a water environment, as well as of possible thermal degradation of sugars during drying and the formation of Maillard reaction products with caramelization of sugars [6]. It may also be influenced by the need to rehydrate the biomass prior to the hydrolysis process. In the dried biomass matrix, the enzyme does not have equal access to macromolecular structures. Therefore, to study yeast multiplication we used the fresh sugar beet pulp after enzymatic treatment with the highest content of sugars.

The metabolic capabilities of *Scheffersomyces stipitis* yeasts allow the consumption of a wide spectrum of sugars, both hexoses and pentoses [25]. Good multiplication was also obtained for other yeast strains belonging to *Candida* sp. and *Saccharomyces* sp., which are often used to obtain SCP from various types of waste plant biomass [26,27]. Other non-*Saccharomyces* yeasts may also have potential for the enrichment of various food wastes. For example, in addition to *Saccharomyces*, mono or co-cultures of *Yarrowia, Metschnikowia, and Kluyveromyces* genera have been used for SCP or lipid production [28,29,30,31,32]. The results for dried sugar beet pulp were less unambiguous, and did not depend on the dose of the enzyme used for hydrolysis. It seems that the reason lies in the earlier heat treatment of the sugar beet pulp and the formation of growth-inhibiting compounds during the drying process. Oven dried sugar beet powders usually exhibit the darkest color, due to Maillard’s reactions [33]. Therefore, both enzyme activity and yeast growth could be more differentiated.

In a study by Yan et al., an engineered strain of *Yarrowia lipolytica* growing on a sugarcane molasses-based medium produced an SCP yield of 151.2 g/L [34]. In another study using the same strain, two-step fermentation of food waste enabled a protein yield of up to 38.8% dry weight [30]. *Metschnikowia pulcherrima* has been shown to rapidly adapt to substrates with high sugar and protein concentrations. In combination with pre-hydrolysis, it produces a high level of protein increment [22]. *Scheffersomyces stipitis* yeast is used for ethanol production, and is metabolically predisposed to high tolerance to alcohol. Protein biosynthesis is a secondary function, in relation to ethanol production, which is reflected in our results [35,36]. The yeast *Kluyveromyces marxianus* has a wide range of applications in single-cell protein production. It has the ability to grow on a variety of culture media, based on many types of sugars. Based on literature data, it grows on fruit, cereal, brewery waste, and dairy waste, yielding a high protein content in the final product (up to 51% DM) [37,38,39]. *Candida utilis* cultured on rice produced an increase in protein of around 1.6% [40]. Despite the use of a different culture medium, this was similar to the values obtained in our study. *Saccharomyces bayanus* and *Saccharomyces cerevisiae* are related yeast strains with similar metabolic capabilities. According to Razzaq et al., fermentation with *S. cerevisiae* leads to protein yields of 5.6% *w*/*v* [41]. According to the literature, *S. bayanus* is used in the wine industry for the assimilation of monosaccharides. It can also be cultured on agriculture waste media, such as rapeseed meal after oil extraction [22,42].

Beet pulp is a source of various nitrogen compounds, although their content may vary. For example, in one study significantly higher protein was produced from oven-dried sugar beet pulp [33]. The reason may be associated with the preparation methods used prior to the drying process. During fermentation, nitrogenous substrates may be assimilated for the production of structural proteins (yeast cells) and functional proteins (enzymes). Some literature studies were taken into account in selecting inorganic nitrogen sources included in the medium for growth and enzyme formation. Considering the published data, the (NH_4_)_2_SO_4_ supplementation was provided [43]. As the amount of best assimilable source of nitrogen did not exceed 0.3% *w*/*v*, it should be completely assimilated by yeast, especially since the cultures lasted relatively a long time, 48 h [44].Some of the protein compounds may come from enzymatic preparations used for the initial hydrolysis of plant waste biomass [7]. Therefore, FAN content is an interesting parameter for evaluating both the quality of waste plant material after yeast revalorization and the ability of yeasts to assimilate low-molecular nitrogen compounds. Free amino nitrogen is an important element in yeast fermentation processes. Yeast metabolism is influenced by the chemical composition of assimilable nitrogen. Ammonium is known to be preferentially assimilated by yeasts and can be the sole assimilable nitrogen source used to complete fermentation. However, FAN can lead to higher maximum fermentation rates when present in combination with ammonium [45].

These results show that the process of enzymatic hydrolysis followed by fermentation allows for the effective reduction of crude fiber content in waste sugar beet pulp, both fresh and dried. Reducing the dosage of the enzyme preparation results in a worsening of the fiber decomposition process and an increase in the fiber content of the biomass. The reduction in crude fiber depends on the tested strain, which may be due to the variability of the hydrolytic capacity of the yeasts. The tested yeasts do not show the ability to metabolize cellulose or lignin, but the products of enzymatic hydrolysis can be assimilated. For example, *Yarrowia lipolytica* is characterized by adaptation to growth in oil materials with a high protein content, thanks to its ability to synthesize lipases, proteases, and peptidases [46]. However, it does not show the ability to hydrolyze polysaccharides, which significantly affects the efficiency of biotechnological processes using beet pulp and the content of final fiber [47]. The situation is different in the case of the yeast *Scheffersomyces stipitis*, which is widely used in the production of bioethanol from lignocellulosic wastes [48,49]. This yeast is able to assimilate pentoses, which is rare in yeasts [50,51]. The combination of the yeast’s enzymatic capacity and enzymatic hydrolysis allows it to grow on a beet pulp-based medium and reduce the crude fiber content [52]. In turn, *Candida utilis* is capable of efficient growth on beet pulp hydrolysate, including in co-cultures [53]. It is worth noting that genera of non-conventional yeasts such as *Yarrowia*, *Kluyveromyces*, and *Metschnikowia* can be considered as probiotic yeasts stimulating the functioning of the intestinal microbiota of animals [54,55].

Depending on the metabolic capacity and mechanism of an animal’s digestive system, a particular feed can only be used by a specific group of livestock. A high fiber content in biomass completely excludes its use as a feed for poultry. An increase in the fiber content of feed results in low digestibility and anti-nutritional properties [56]. For cattle and pigs, it is acceptable to include fiber components in the feed and combine them with other nutritional elements [57,58]. There are two main types of feed in mixed animal feeds: concentrated feed and roughage feed. Concentrated feeds are characterized by a high content of easily absorbable carbohydrates and a rich protein profile. Roughages have reduced nutritional value and increased fiber content [59,60]. Unprocessed sugar beet pulp is categorized as roughage, but through the simple fermentation and biosynthesis of single cell protein it is possible to improve the nutritional value of the biomass and categorize it as nutritious feed, due to the increase in digestible protein [61].

## 4. Materials and Methods

### 4.1. Research Material

The research material was sugar beet pulp after the process of white sugar production. This waste material was supplied in the two forms: fresh sugar beet pulp from the Sugar Factory in Dobrzelin, Poland and dried sugar beet pulp from the Sugar Factory in Werbkowice, Poland. Both companies are part of the National Food Industry Group, Poland.

### 4.2. Yeasts Strains

The yeast strains used during the cultivation are listed in Table 4.

### 4.3. Enzyme Preparations

To conduct enzymatic hydrolysis of waste biomass, two types of enzymatic preparation were used: Viscozyme^®^ L by Novozyme and UltraFlo^®^ Max by Novozyme. The preparations used are industrial products with established homogeneous compositions containing the following enzymes: cellulase, xylanase, pectinase, invertase [17,62]. The enzymatic activities of the used preparations are presented in Table 5.

### 4.4. Sample Preparation

A portion of 40 g of fresh sugar beet pulp was placed in a conical flask and 60 mL of distilled water was added to obtain a total sample mass of 100 g. In the case of dried pulp, the portion of biomass was altered to achieve a similar dry mass (DM) content. To this end, 10 g of dried sugar beet pulp was placed in a flask and 90 mL of distilled water was added. The samples were sterilized at 121 °C for 15 min to remove any pathogenic microorganisms. After cooling the samples, pre-hydrolysis was conducted using enzyme preparations. The following enzyme doses were used: 0.5 mL/10 g DM; 0.25 mL/10 g DM; 0.125 mL/10 g DM (calculated enzyme units are presented in Appendix A). The samples were incubated at 50 °C for 4 h.

### 4.5. Simultaneous Sachcrification and Fermentation (SSF)

The enzyme preparations were not inactivated during pre-hydrolysis, so bioconversion was carried out through simultaneous saccharification and fermentation (SSF). After pre-hydrolysis, each sample was supplemented with ammonium sulphate as an additional nitrogen source, at a dose of 0.3 g/100 mL. Following supplementation, the samples were inoculated with selected yeast strains and cultured on an orbital shaker at 210 rpm for 48 h at ambient temperature (approximately 21 °C). The suspensions were standardized using an optical densitometer to range of 2–3 McF scale. After inoculation, the initial numbers of yeast cells in the hydrolysates were determined using the classical plate count method.

### 4.6. Carbohydrates Determinantion

After pre-hydrolysis, the efficiency of the procedure was checked by analyzing the sugar content. Four types of carbohydrates were selected for further analysis as the main carbon sources for yeast growth: D-xylose, D-mannose, D-fructose, and D-glucose. Carbohydrates were determined using the Megazyme kit (K-XYLOSE; K-MANGL) in accordance with the manufacturer’s specifications.

### 4.7. Microbial Growth

Yeast growth was monitored using the classical plate count method, with YGC agar medium (yeast extract 5 g, glucose 20 g, chloramphenicol 0.1 g, agar 15 g per 1 L). Each sample after yeast multiplication, dilution, and inoculation was incubated for 48 h at 30 °C. Inoculated samples without cultivation were used as control samples.

### 4.8. Protein Content Determination

The Kjeldahl method was used to determine the increase in protein content of the biomass in the fermented samples. For this purpose, biomass was separated from post culture liquid by centrifugation. The analytical sample was transferred to a flask (1 g) and filled with concentrated sulphuric acid (15 mL). The sample was heated at 550 °C in a SpeedDigester K-425 by Büchi until a colorless solution was obtained. The sample was placed in KjelFlex K-360 apparatus, diluted with distilled water (40 mL), then neutralized with a 30% NaOH solution (60 mL), steam distilled into 2% boric acid solution (40 mL), and titrated (TitroLine^®^5000 by SI Analytics) with a standard hydrochloric acid 0.1 mol/L solution. Non-fermented, non-hydrolyzed samples were used as a reference. During calculations the influence of exogenous enzymes protein was also excluded.

### 4.9. Free Amino Nitrogen (FAN) Content Determination

The free amino nitrogen content in the yeast post-culture liquid was determined after centrifugation. The ninhydrin method, was used, as described in the protocol from Eppendorf [63]. The diluted post-culture liquid sample was placed in a test tube, to which distilled water and ninhydrin reagent were added. The sample was incubated at 100 °C for 16 min. After cooling the sample to room temperature, the dilution solution was added, and the absorbance was measured at a wavelength of 570 nm. Unfermented samples after hydrolysis were used as a control.

### 4.10. Determination of Crude Fiber Content

A FOSS Fibertec^®^ 8000 apparatus was used to determine changes in the crude fiber content of the tested biomass. The biomass samples were placed in a crucible prepared previously. The samples were weighed with an accuracy of 0.1 mg. The samples were washed with acetone to remove any potential fats. The crucibles were then placed in the apparatus and hot extraction was performed successively in a solution of 1.25% sulphuric acid and 1.25% potassium hydroxide solution (in 150 mL of each solution for 30 min). After extraction, the samples were washed three times in acetone, then placed in a drying oven at 130 °C for 2 h. After drying, the samples were cooled to ambient temperature and weighed to the nearest 0.1 mg. The samples were ashed at 525 °C for 3 h. Finally, they were cooled to ambient temperature and weighed. The results were used to derive the crude fiber content of the sample according to the formula
Crude fiber [%] = (W_2_ − W_3-C_) ÷ W_1_ × 100,
where: W_1_—sample mass; W_2_—crucible mass after extraction; W_3_—crucible mass after ashing; _C_—control sample [64].

### 4.11. Statistical Analysis

Statistical analysis was performed by analysis of variance (one-way ANOVA) at a significance level *p* ≤ 0.05 using STATISTICA 13.1 (StatSoft, Tulsa, OK, USA) to specify differences. Post-hoc analysis was performed if statistical difference was detected (Tukey’s test, significance *p* ≤ 0.05).

## 5. Conclusions

This study demonstrates that simultaneous hydrolysis and fermentation can be used to revalorize waste sugar beet pulp. Enzymatic degradation of polysaccharides releases simple sugars, which are assimilated by both conventional and non-conventional yeasts. All the tested yeast strains were capable of growing on the waste plant biomass hydrolysates. As a result, cell multiplication of 10^7^–10^8^ CFU/mL was obtained, resulting in a significant increase in protein content. The highest protein increases were recorded for the fodder yeast *Candida utilis* (2.33%) and xylose-fermenting yeast *Scheffersomyces stipitis* (3.04%), cultivated on fresh and dried sugar beet pulp hydrolysates, respectively. The most substantial reduction in crude fiber was obtained for the processes conducted with *Saccharomyces cerevisiae* Ethanol Red in fresh biomass and with *Candida utilis* in dried biomass hydrolysates. The results of this study provide a basis for further research on the use of sugar beet pulp biomass for the production of fodder, contributing to the goals of sustainable sugar production. Future work should consider the possibility of obtaining preparations both enriched in protein and characterized by a lower content of fiber. This would make it possible to increase the share of SCP based on sugar beet pulp in compound feed, especially for non-ruminant animals.

## Figures and Tables

**Figure 1 molecules-28-02064-f001:**
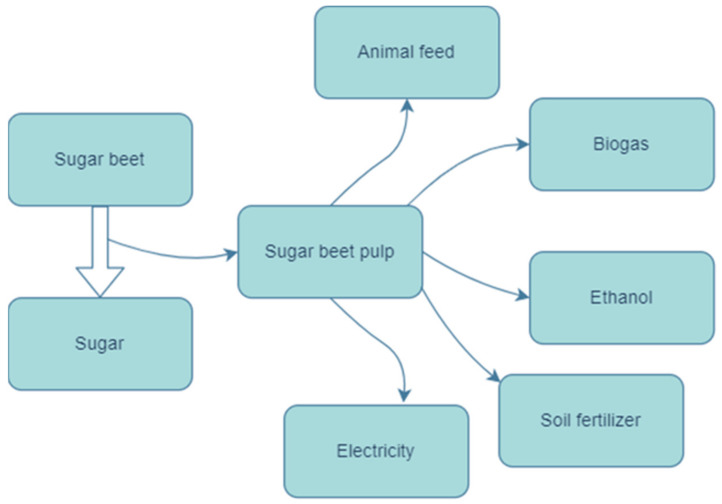
Potential uses of sugar beet pulp waste biomass by biorefinery conglomerates [6,7].

**Figure 2 molecules-28-02064-f002:**
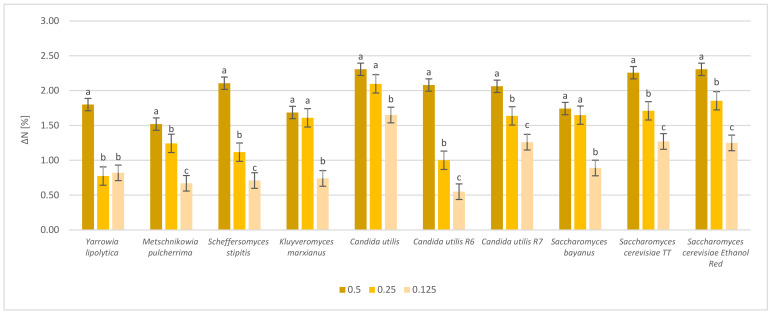
Protein content increase, expressed as ΔN [%], after cultivation of fresh sugar beet pulp with selected yeast strains. The fresh biomass was hydrolyzed with various enzyme doses (0.5 mL/10 g DM; 0.25 mL/10 g DM; 0.125 mL/10 g DM). The results were adjusted to exclude the influence of enzyme preparations. a, b, c—indicators of statistically significant difference, mean values for strain with different letters are significantly different (*p* < 0.05).

**Figure 3 molecules-28-02064-f003:**
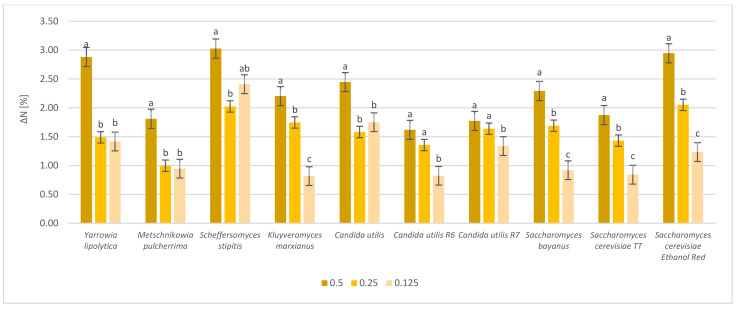
Protein content increase expressed as ΔN [%], after cultivation of dried sugar beet pulp with selected yeast strains. The dried biomass was hydrolyzed with various enzyme doses (0.5 mL/g DM; 0.25 mL/10 g DM; 0.125 mL/10 g DM). The results were adjusted to exclude influence of enzyme preparations. a, b, c—indicators of statistically significant difference, mean values for strain with different letters are significantly different (*p* < 0.05).

**Figure 4 molecules-28-02064-f004:**
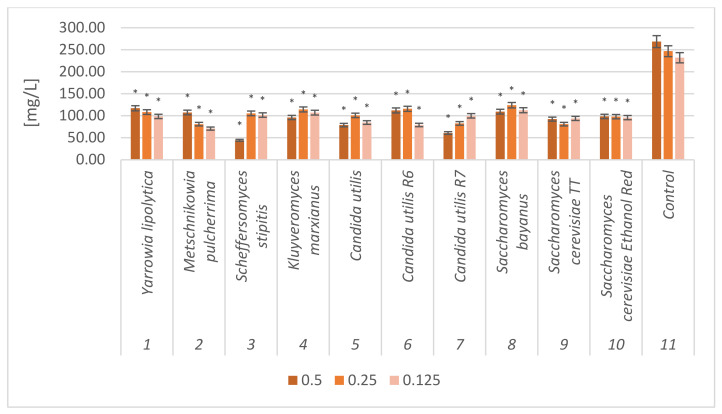
Free amino nitrogen (FAN) content in fresh sugar beet pulp samples after yeast cultivation. The biomass was hydrolyzed with different enzyme concentrations (0.5 mL/10 g DM; 0.25 mL/10 g DM; 0.125 mL/10 g DM); *—indicator of statistical difference relative to the control sample (*p* < 0.05).

**Figure 5 molecules-28-02064-f005:**
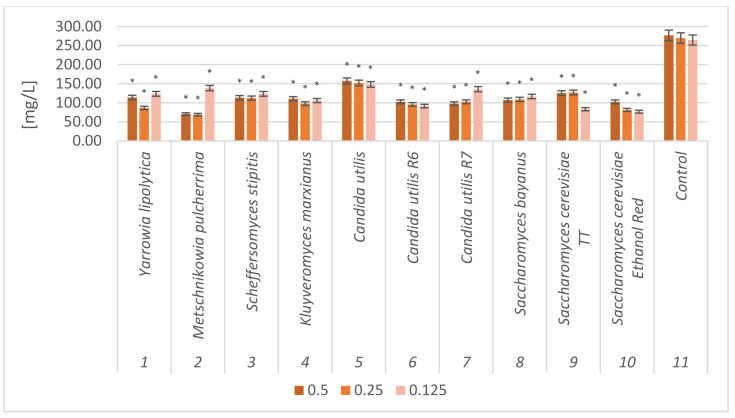
Free amino nitrogen (FAN) content in dried sugar beet pulp samples after yeast cultivation. The biomass was hydrolyzed with different enzyme concentrations (0.5 mL/10 g DM; 0.25 mL/10 g DM; 0.125 mL/10 g DM). *—indicator of statistical difference relative to the control sample (*p* < 0.05).

**Figure 6 molecules-28-02064-f006:**
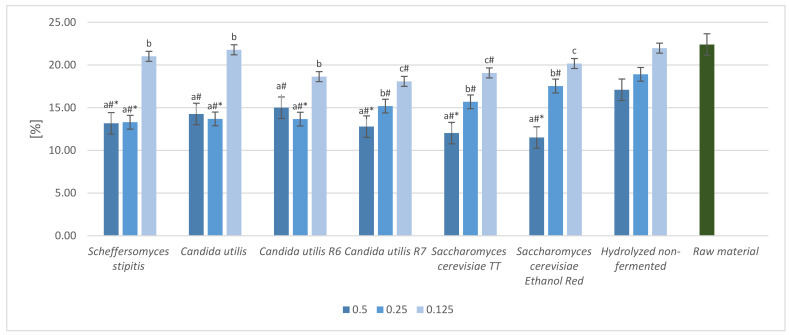
Crude fiber (CF) content after pre-hydrolysis and yeast cultivation on fresh sugar beet pulp hydrolyzed with different enzyme doses (0.5 mL/10 g DM; 0.25 mL/10 g DM; 0.125 mL/10 g DM). a, b, c—indicators of statistically significant difference, mean values for strain with different letters are significantly different (*p* < 0.05). *—indicator of significant difference relative to mean value of hydrolyzed non-fermented sample, #—indicator of significant difference relative to the mean value for the raw material (*p* < 0.05).

**Figure 7 molecules-28-02064-f007:**
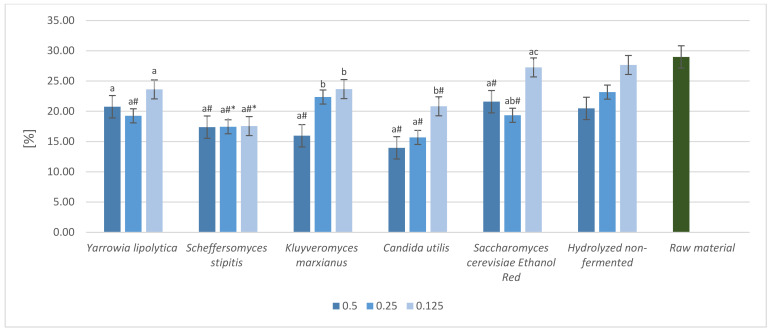
Crude Fiber (CF) content after pre-hydrolysis and yeast cultivation with dried sugar beet pulp hydrolyzed with different enzyme doses (0.5 mL/10 g DM; 0.25 mL/10 g DM; 0.125 mL/10 g DM). a, b, c—indicators of statistically significant difference, mean values for strain with different letters are significantly different (*p* < 0.05). *—indicator of significant difference relative to mean value of hydrolyzed non-fermented sample, #—indicator of significant difference relative to the mean value for the raw material (*p* < 0.05).

**Table 1 molecules-28-02064-t001:** Sugar profiles in tested waste biomass before and after enzymatic hydrolysis [g/L] with various enzyme concentrations.

Carbohydrate	Fresh Sugar Beet Pulp				Dried Sugar Beet Pulp			
	Enzymatic Treatment [mL/10 g DM]				Enzymatic Treatment [mL/10 g DM]			
	0 (Without)	0.5	0.25	0.125	0 (Without)	0.5	0.25	0.125
D-Xylose	0.002 ± 0.001	1.113 ± 0.032	0.856 ± 0.031	0.826 ± 0.045	0.013 ± 0.003	1.046 ± 0.045	0.761 ± 0.050	0.681 ± 0.009
D-Mannose	0.011 ± 0.004	0.649 ± 0.046	0.088 ± 0.012	0.013 ± 0.003	0.021 ± 0.006	0.431 ± 0.020	0.261 ± 0.010	0.201 ± 0.021
D-Fructose	0.044 ± 0.010	0.655 ± 0.027	0.081 ± 0.008	0.587 ± 0.065	0.051 ± 0.011	0.564 ± 0.016	0.201 ± 0.031	0.164 ± 0.043
D-Glucose	0.049 ± 0.009	16.823 ± 0.097	14.277 ± 0.056	13.217 ± 0.080	0.059 ± 0.008	14.374 ± 0.080	13.031 ± 0.094	11.264 ± 0.032

**Table 2 molecules-28-02064-t002:** Yeast growth [CFU/mL] in a culture medium with fresh sugar beet pulp pretreated by enzymatic hydrolysis with various enzyme doses A–C [ml/10 g of dry mass].

Yeast Strain	0.5	0.25	0.125	Control
*Yarrowia lipolytica*	AV: 1.54 × 10^7 a#^	AV: 1.32 × 10^7 a^	AV: 1.90 × 10^7 a#^	AV: 6.50 × 10^6^
	SD: 5.90 × 10^6^	SD: 4.60 × 10^6^	SD: 4.00 × 10^6^	SD: 1.29 × 10^6^
*Metschnikowia pulcherrima*	AV: 2.46 × 10^7 ab^	AV: 2.38 × 10^7 b^	AV: 4.66 × 10^7 a#^	AV: 7.05 × 10^6^
	SD: 3.58 × 10^6^	SD: 4.49 × 10^6^	SD: 2.34 × 10^7^	SD: 6.86 × 10^5^
*Scheffersomyces stipitis*	AV: 2.82 × 10^8 a#^	AV: 1.13 × 10^8 ab^	AV: 3.32 × 10^7 b^	AV: 3.18 × 10^7^
	SD: 1.31 × 10^8^	SD: 3.22 × 10^7^	SD: 1.64 × 10^7^	SD: 8.88 × 10^6^
*Kluyveromyces marxianus*	AV: 4.82 × 10^7 a#^	AV: 2.10 × 10^7 b^	AV: 2.52 × 10^7 b^	AV: 1.34 × 10^7^
	SD: 8.11 × 10^6^	SD: 3.67 × 10^6^	SD: 9.56 × 10^6^	SD: 7.33 × 10^6^
*Candida utilis*	AV: 2.23 × 10^8 a#^	AV: 2.01 × 10^8 a#^	AV: 2.60 × 10^8 a#^	AV: 2.43 × 10^7^
	SD: 8.54 × 10^7^	SD: 4.10 × 10^7^	SD: 6.39 × 10^7^	SD: 1.69 × 10^7^
*Candida utilis* R6	AV: 3.74 × 10^7 ab^	AV: 5.78 × 10^7 a#^	AV: 2.26 × 10^7 b^	AV: 2.34 × 10^7^
	SD: 6.91 × 10^6^	SD: 2.29 × 10^7^	SD: 1.10 × 10^7^	SD: 5.29 × 10^6^
*Candida utilis* R7	AV: 4.12 × 10^7 a^	AV: 4.78 × 10^7 a^	AV: 2.13 × 10^8 b#^	AV: 5.95 × 10^7^
	SD: 7.26 × 10^6^	SD: 4.79 × 10^7^	SD: 5.90 × 10^7^	SD: 2.14 × 10^7^
*Saccharomyces bayanus* BC S103	AV: 1.34 × 10^8 a#^	AV: 1.63 × 10^8 a#^	AV: 1.60 × 10^8 a#^	AV: 3.56 × 10^7^
	SD: 2.08 × 10^7^	SD: 2.52 × 10^7^	SD: 5.51 × 10^7^	SD: 1.11 × 10^7^
*Saccharomyces cerevisiae* TT	AV: 1.07 × 10^8 ab^	AV: 5.53 × 10^7 a^	AV: 1.75 × 10^8 b#^	AV: 7.23 × 10^7^
	SD: 2.69 × 10^7^	SD: 5.21 × 10^6^	SD: 8.73 × 10^7^	SD: 1.74 × 10^7^
*Saccharomyces cerevisiae* Ethanol Red	AV: 2.16 × 10^8 a#^	AV: 1.87 × 10^8 a#^	AV: 2.05 × 10^8 a#^	AV: 8.08 × 10^7^
	SD: 2.29 × 10^7^	SD: 1.18 × 10^7^	SD: 1.64 × 10^7^	SD: 9.43 × 10^6^

a, b—indicators of statistically significant difference, mean values for strain with different letters are significantly different (*p* < 0.05). #—indicator of significant difference from the control mean value (*p* < 0.05).

**Table 3 molecules-28-02064-t003:** Yeast growth [CFU/mL] in culture medium with dried sugar beet pulp pretreated by enzymatic hydrolysis with various enzyme doses D–F [ml/10 g DM].

Yeast Strain	0.5	0.25	0.125	Control
*Yarrowia lipolytica*	AV: 1.36 × 10^8 a#^	AV: 1.63 × 10^8 a#^	AV: 1.38 × 10^8 a#^	AV: 2.46 × 10^7^
	SD: 2.93 × 10^7^	SD: 5.94 × 10^7^	SD: 4.64 × 10^7^	SD: 3.92 × 10^6^
*Metschnikowia pulcherrima*	AV: 5.46 × 10^7 a^	AV: 2.88 × 10^7 ab^	AV: 1.20 × 10^8 b#^	AV: 2.46 × 10^7^
	SD: 1.13 × 10^7^	SD: 5.84 × 10^6^	SD: 2.40 × 10^7^	SD: 2.08 × 10^6^
*Scheffersomyces stipitis*	AV: 9.34 × 10^7 a#^	AV: 6.68 × 10^7 ab#^	AV: 1.26 × 10^8 b#^	AV: 3.38 × 10^7^
	SD: 1.06 × 10^7^	SD: 9.04 × 10^6^	SD: 1.28 × 10^7^	SD: 1.74 × 10^7^
*Kluyveromyces marxianus*	AV: 1.20 × 10^8 a#^	AV: 5.36 × 10^7 b^	AV: 5.42 × 10^7 b^	AV: 1.22 × 10^7^
	SD: 2.00 × 10^7^	SD: 3.86 × 10^7^	SD: 1.22 × 10^7^	SD: 1.29 × 10^6^
*Candida utilis*	AV: 1.10 × 10^8 a^	AV: 1.12 × 10^8 a^	AV: 1.62 × 10^8 a^	AV: 1.37 × 10^8^
	SD: 1.60 × 10^7^	SD: 1.84 × 10^7^	SD: 5.44 × 10^7^	SD: 1.74 × 10^7^
*Candida utilis* R6	AV: 2.14 × 10^7 a^	AV: 2.40 × 10^7 a#^	AV: 1.16 × 10^7 a#^	AV: 2.75 × 10^6^
	SD: 4.48 × 10^6^	SD: 9.20 × 10^6^	SD: 1.52 × 10^6^	SD: 1.00 × 10^6^
*Candida utilis* R7	AV: 5.67 × 10^7 a^	AV: 4.96 × 10^7 b^	AV: 2.80 × 10^7 c^	AV: 3.80 × 10^7^
	SD: 2.89 × 10^6^	SD: 1.23 × 10^7^	SD: 9.60 × 10^6^	SD: 3.89 × 10^6^
*Saccharomyces bayanus* BC S103	AV: 1.23 × 10^8 a#^	AV: 1.36 × 10^8 b#^	AV: 1.26 × 10^8 b#^	AV: 1.27 × 10^7^
	SD: 1.34 × 10^7^	SD: 1.68 × 10^7^	SD: 1.52 × 10^7^	SD: 2.26 × 10^6^
*Saccharomyces cerevisiae* TT	AV: 4.30 × 10^7 a^	AV: 1.33 × 10^8 b#^	AV: 2.04 × 10^8 b#^	AV: 1.14 × 10^7^
	SD: 1.40 × 10^7^	SD: 2.98 × 10^7^	SD: 6.48 × 10^7^	SD: 6.25 × 10^6^
*Saccharomyces cerevisiae* Ethanol Red	AV: 1.34 × 10^8 a#^	AV: 1.64 × 10^8 ab#^	AV: 1.92 × 10^8 b#^	AV: 3.54 × 10^7^
	SD: 1.52 × 10^7^	SD: 2.72 × 10^7^	SD: 3.04 × 10^7^	SD: 7.68 × 10^6^

a, b, c—indicators of statistically significant difference, mean values for strain with different letters are significantly different (*p* < 0.05). #—indicator of significant difference from the control mean value (*p* < 0.05).

**Table 4 molecules-28-02064-t004:** Yeast strains used in the study.

Yeast Strain	Strain Code
*Yarrowia lipolytica*	LOCK 0264
*Metschnikowia pulcherrima*	NCYC 747
*Scheffersomyces stipitis*	NCYC 1541
*Kluyveromyces marxianus*	LOCK 0024
*Candida utilis*	LOCK 0021
*Candida utilis*	R6
*Candida utilis*	R7
*Saccharomyces bayanus* BC S103	Fermentis Lesaffre for Beverages
*Saccharomyces cerevisiae* TT	LOCK 0105
*Saccharomyces cerevisiae* Ethanol Red	Leaf/Lesaffre Advanced Fermentation

**Table 5 molecules-28-02064-t005:** Activities of enzymes used in the studies [U/mL].

Enzyme Preparation	Cellulase	Invertase	Xylanase	Pectinase
Viscozyme	20.9	61.2	25.9	312.6
Ultraflo Max	32.7	1.8	64.7	21.2

## Data Availability

Not applicable.

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
