# Peer review of "Sugar Beet Pulp as a Biorefinery Substrate for Designing Feed"

_molecules, 2023, doi:10.3390/molecules28052064_

Round 1

Reviewer 1 Report

The authors assessed sugar beet pulp as carbon source to growth yeasts, after enzymatic hydrolysis. After read the manuscript, I did not find the connection between the conclusion and objective of the research. In this sense, there are not a methodology to design a new culture medium based on sugar beet pulp or a methodology to explain how the authors selected one or several yeasts compatible with the carbon source. Is yeast extract the only nitrogen source present in YGC medium? Is this compatible with the Sustainable Development Goals? About format, my suggestion is to improve all the figures with high quality images, avoid Excel graphs. My general recommendation is minor revisions. Some specific comments are listed below:

Figure 1. This figure was not cited in the manuscript. Why is an arrow inside another arrow? please revise.

Table 1. The second column says "the origin"... I suggest to replace by Strain code, or similar.

Line 88-89, please mention the concentration for each component of YGC medium.

Line 253. This is an engineered strain, please clarify it in the manuscript.

Author Response

Dear Reviewer,

Below we provide answers to the issues raised in the review. We hope that we meet your expectations regarding the corrections and quality of the article.

Sincerely

Authors

In this sense, there are not a methodology to design a new culture medium based on sugar beet pulp or a methodology to explain how the authors selected one or several yeasts compatible with the carbon source.

  • Additional explanation for yeast strains choice has been incorporated in the text - Lines 204-207. The methodology of hydrolysates preparation has been improved and enclosed in the paragraph 2.5, lines 104-107.

Is yeast extract the only nitrogen source present in YGC medium?

  • YGC medium was used for plate count method. During cultivation, hydrolyzed sugar beet pulp and added ammonium sulfate were sources of nitrogen for yeast growth.

Is this compatible with the Sustainable Development Goals?

  • Valuable feed can be obtained, contributing to the 12th Sustainable Development Goal considering the sustainable consumption and production, especially in terms of sustainable management and efficient use of natural resources. This information has been incorporated in the text lines 64-67

Figure 1. This figure was not cited in the manuscript. Why is an arrow inside another arrow? please revise.

  • Citation has been added and graphic was corrected.

Table 1. The second column says "the origin"... I suggest to replace by Strain code, or similar.

  • Name was replaced

Line 88-89, please mention the concentration for each component of YGC medium.

  • Concentration of each ingredient has been added.

Line 253. This is an engineered strain, please clarify it in the manuscript.

  • Information has been added.

Reviewer 2 Report

The article molecules-2208885 suggests cultivation with yeast to increase the protein content and reduce the cru fiber content of sugar beet pulp. However, the adopted methodology is inappropriate. Several analytical procedures were conducted inappropriately. I suggest rejecting this article.

Introduction

What is done currently with sugar beet pulp in Poland and other countries? This information cannot be omitted;

- Line 27: sugarcane is the main raw material for world sugar production. Therefore, the main residues of the sugar industry are sugarcane bagasse and straw. The generation of these two residues is much greater than that of sugar beet pulp;

- A paragraph was missing on the importance of enzymatic hydrolysis prior to or simultaneous to fermentation;

- Why this work opted for protein enrichment of the pulp instead of the other potential uses shown in Figure 1? In summary: the justification for the work is weak;

Experimental

- Line 54: this methodology must not be omitted;

- Line 57: Change “analyses” to “cultivations”;

- Lines 60-62: What enzymes are present in this formulation? Such information can help to understand why there was a higher concentration of glucose. In addition, it is necessary to at least cite articles that reported the composition of sugar beet pulp;

- Topic 2.4: it should be divided into 2: sample preparation and simultaneous hydrolysis and fermentation;

- Lines 71-72: Enzymes should be added based on enzyme activity and not on the basis of cocktail volume. Enzymes are denatured due to storage. This was a very serious error, which demonstrates a lack of knowledge in handling enzymes;

- Lines 76-77: What was the concentration of inoculated yeast cells? Was the cell concentration similar for the different yeasts? This omission was another serious mistake;

- Line 95: “analytical biomass sample”? This term is inappropriate;

- Line 103: One more experimental procedure that was omitted by the authors. What were the volumes of each solution used in the test?

- Lines 114-115: What was the extraction time adopted? This experimental procedure, as well as the others, should be referenced;

Topic 3.3: The authors did not mention or did not perform a control to discount the inoculated cells, not even the enzymes that were added, which interfere in the determination of N by the Kjeldahl method. This can compromise the results;

Topics 3.5: For the determination of crude fiber, the chosen control was inappropriate. The authors should have used the controls adopted in the determination of FAN, as Viscozyme has cellulases that would reduce the cellulose content in the pulp;

Line 378: Many of the conclusions are compromised due to errors in conducting the experiments.

Author Response

Dear Reviewer,

Below we provide answers to the issues raised in the review. We hope that we meet your expectations regarding the corrections and quality of the article.

Sincerely

Authors

What is done currently with sugar beet pulp in Poland and other countries? This information cannot be omitted;

  • Additional information about current situation in Poland included in the text in Introduction. Lines: 35-38, 52-57

Line 27: sugarcane is the main raw material for world sugar production. Therefore, the main residues of the sugar industry are sugarcane bagasse and straw. The generation of these two residues is much greater than that of sugar beet pulp;

  • Due to climate conditions, sugarcane is impossible to effectively cultivate in Europe. There, commonly used plant in Europe is sugar beet. Full explanation has been included in the text, Lines: 27-29, 33-36

A paragraph was missing on the importance of enzymatic hydrolysis prior to or simultaneous to fermentation;

  • A new paragraph with details concerning the function and advantages o simultaneous saccharification and fermentation was added in introduction, Lines: 57-62

Why this work opted for protein enrichment of the pulp instead of the other potential uses shown in Figure 1? In summary: the justification for the work is weak;

  • More details concerning the justification was added in introduction, Lines: 47-51, 64-69

Line 54: this methodology must not be omitted;

  • This sentence was placed by mistake, therefore it was removed.

Line 57: Change “analyses” to “cultivations”;

  • Word has been changed.

Lines 60-62: What enzymes are present in this formulation? Such information can help to understand why there was a higher concentration of glucose. In addition, it is necessary to at least cite articles that reported the composition of sugar beet pulp;

  • Specific enzymes in this formulation has been added. Specific enzymatic activity can be seen in Table 1 and Table S1.

Topic 2.4: it should be divided into 2: sample preparation and simultaneous hydrolysis and fermentation;

  • Description divided into 2 sections, as recommended.

Lines 71-72: Enzymes should be added based on enzyme activity and not on the basis of cocktail volume. Enzymes are denatured due to storage. This was a very serious error, which demonstrates a lack of knowledge in handling enzymes;

  • Calculated enzymatic activity included in Table 2 and in supplementary Table S1.

Lines 76-77: What was the concentration of inoculated yeast cells? Was the cell concentration similar for the different yeasts? This omission was another serious mistake;

  • The information about the standardization of yeast suspensions used for the culture inoculation was added (Lines 109-112). It was based on optical density of the suspension. The process was followed by CFU assay at the beginning of the fermentation - 0h. Therefore, the initial cell concentration was known for the inoculated medium.

Line 95: “analytical biomass sample”? This term is inappropriate;

  • Line: 129

Line 103: One more experimental procedure that was omitted by the authors. What were the volumes of each solution used in the test?

  • Full protocol with step-by-step procedure can be seen in cited document No 20.

Lines 114-115: What was the extraction time adopted? This experimental procedure, as well as the others, should be referenced;

  • Required methodological details have been added and reference attached. Lines: 153. Reference No 21

Topic 3.3: The authors did not mention or did not perform a control to discount the inoculated cells, not even the enzymes that were added, which interfere in the determination of N by the Kjeldahl method. This can compromise the results;

  • We provided an additional analyzes to calculate and exclude an influence of enzyme preparations (Lines 135-136, Figures 2 and 3), however the discount connected with the inoculated cells was neglected. Instead of this authors confirmed the increase of the number of yeast cell during the cultivation.

Topics 3.5: For the determination of crude fiber, the chosen control was inappropriate. The authors should have used the controls adopted in the determination of FAN, as Viscozyme has cellulases that would reduce the cellulose content in the pulp;

  • We carried out additional tests to include hydrolyzes non-fermented sample as additional control sample (Figures 6 and 7).

Line 378: Many of the conclusions are compromised due to errors in conducting the experiments.

  • The conclusions were rephrased based on the outcomes taking into account the new analyses.

Round 2

Reviewer 2 Report

The article molecules-2208885 presented improvements, but other corrections are necessary:

Line 106: Only now did the authors mention ammonium sulfate supplementation. How did the authors discount the ammonium sulfate not assimilated by the yeasts in the determination of N by the Kjeldahl method?

Line 134: change “N” to “mol/L”;

Lines 135-136: Authors should mention that such discounted enzymes are exogenous, as there are enzymes produced in vivo in the cultivations.

Author Response

Dear Reviewer

Thank you for your comments. We have improved the methodological description of our research as suggested. An explanation to your comments is provided below.

Authors

Line 106: Only now did the authors mention ammonium sulfate supplementation.

Of course, the nitrogen source used in the production medium is one of the main factors affecting growth and enzyme production and levels. Similar literature studies were taken into account the selecting inorganic nitrogen sources included in the medium for growth and enzyme formation, e.g. the addition of NH4NO3, NaNO3, and (NH4)2SO4. The best results were obtained with (NH4)2SO4 supplementation [doi: 10.1016/j.ibiod.2004.09.001].

Synthetic laboratory media are usually supplemented with ammonium salts (mainly ammonium sulfate) to provide the necessary nitrogen. In turn, complex media, e.g., molasses, which are naturally rich in nitrogen-containing compounds, do not require additional nitrogen supplementation. Other sources of nitrogen include yeast/malt extract, urea and nitrates.

Therefore, we decided to use (NH4)2SO4 as a nitrogen source in an amount not exceeding 0.3% w/v. It is not a significant amount and, as the best assimilable source of nitrogen, it should be completely assimilated by yeast, especially since the cultures lasted quite a long time, 48 hours. Literature data show that assimilable nitrogen is taken up within the first few days of fermentation processes [doi:10.1016/j.fm.2018.12.002].

Additional text considering this issue has been added - Lines: 336-340.

Line 106: How did the authors discount the ammonium sulfate not assimilated by the yeasts in the determination of N by the Kjeldahl method?

Ammonium sulfate was dissolved in sample (liquid fraction). After the fermentation process, biomass was separated from post-culture liquid by centrifugation. Only solid fraction was tested for protein content. Not assimilated ammonium sulfate was present in liquid fraction. To determine not assimilated ammonium sulfate, we tested post-culture liquid toward free amino nitrogen content. Moreover, the control sample was also separated by centrifugation, non-hydrolyzed, non-fermented biomass. Only the uptake of FAN was discussed.

To clarify this matter the description of methodology has been extended - Lines 129-130, 140

Line 134: change “N” to “mol/L”;

Corrected

Lines 135-136: Authors should mention that such discounted enzymes are exogenous, as there are enzymes produced in vivo in the cultivation.

Information has been added. (Line 137)